Coping self-efficacy and social support as
predictors of adolescent sex trafficking exit:
Results of a secondary analysis. PLoS ONE 19(1):
e0291207. https://doi.org/10.1371/journal.
pone.0291207

PAKISTAN

**Data Availability Statement:** The underlying data
can be found in Supplemental File 4.

**Funding:** This project was supported by a Victims
of Crime Act (VOCA) federal pass-through grant

# Coping self-efficacy and social support as predictors of adolescent sex trafficking exit: Results of a secondary analysis

**Mary K. Twis** [1]*, **Andrea Cimino**[2], **Marilyn Plunk**[1]

1 Department of Social Work, Texas Christian University, Fort Worth, Texas, United States of America,
2 Danger Assessment Training and Technical Assistance Center, Johns Hopkins University, Baltimore,
Maryland, United States of America

* Mary.twis@tcu.edu

## Abstract

### Introduction

Social work case management services are increasingly available to youth who want to exit
commercial sexual exploitation (CSE). However, few empirical studies investigate the effi-
cacy of such services, particularly whether these services promote an exit from CSE. Guided
by ecological systems theory and the Intentions to Exit Prostitution (IEP) model, this study
investigates the efficacy of social work case management services for youth CSE survivors.

### Methods

Youth survivors of CSE ($n = 95$) participated in a one-group, quasi-experimental double pre/
posttest design study. Measures included the Multidimensional Scale of Perceived Social
Support (MSPSS), Coping Self-Efficacy Scale (Cop-SE), and a modified version of the Uni-
versity of Rhode Island Change Assessment (URICA) surveys at zero- and six-months fol-
lowing study commencement. The research team also collected demographic and
victimization data, the number and type of social work case management services received,
and goal plan data. Analyses included repeated measures tests and linear and multinomial
logistic regressions to determine if doses of social work case management are predictive of
the positive short-term outcomes that are linked to increased readiness to exit CSE.

### Results

Youth CSE survivors experienced upward trends in perceived social support and coping
self-efficacy scores between zero- and six-months following study commencement. Linear
and logistic regressions demonstrated that variables like months of service time, trafficking
classification, goal counts, race, and age can predict outcomes like survivor social support,
coping self-efficacy, and intention to change behaviors that can lead to revictimization.

### Implications

Results suggest social work case management services that improve coping self-efficacy
and perceived social support can lead to cognitive changes that promote an exit from CSE.

awarded to the Office of the Texas Governor (OOG) Child Sex Trafficking Team and Traffick911. The sponsor of this protocol – Traffick911 – has assisted with the study design and data collection plan described in this protocol. Traffick911 administrators have approved of the protocol described herein. The original funders of this project, including VOCA and the OOG, had no role in study design, data collection and analysis, decision to publish, or preparation of the manuscript. The opinions, findings, conclusions, and recommendations expressed in this publication are those of the authors and do not necessarily reflect the views of VOCA, OOG, or Traffick911. (www.traffick911.com).

**Competing interests:** The authors have declared that no competing interests exist.

Practitioners should target services that adhere to dimensions of the IEP as these improvements are likely to support positive outcomes for youth survivors of CSE.

## Introduction

The purpose of this study is to investigate the effect of social work case management services on self-efficacy, coping skills, and readiness to exit among youth survivors of commercial sexual exploitation (CSE). The study is guided by the ecological systems theory [1], a framework to understand how the environment shapes behavior, and the Intentions to Exit Prostitution (IEP) model [2–4], a framework to understand cognitive changes associated with an exit from CSE. The specific focus of this study is on the commercial sexual exploitation of youth (CSEY), which occurs when a child or adolescent exchanges sex for money, goods, or services, with or without a third-party facilitator like a trafficker or "pimp" [5, 6]. Federal law considers any commercial sexual exploitation of a minor a form of sex trafficking regardless of the youth's intent [7–12]. The terms CSEY are often used interchangeably with "commercial sexual exploitation of children" (CSEC) or "domestic minor sex trafficking" (DMST). This study uses the term CSEY because, although all of the participants in this study *were* trafficked as minors, some had reached the age of 18 by the time they engaged with aftercare services in the form of social work case management.

It is important to investigate CSEY survivors' service outcomes because revictimization and re-trafficking are common [13] and the hidden nature of the population makes them hard to research [14, 15] resulting in few reliable estimates of CSEY prevalence, and few studies on whether social work case management services effectively promote an exit from CSE. Research suggests that CSEY victims are likely to face several serious repercussions during and following their victimization [16–25], thereby complicating their recovery, assimilation into broader society, and their ability to successfully exit CSE. Literature has well-documented barriers to exiting. For instance, survivors' experiences with trauma, physical violence and threats, and mental health symptoms diminish their self-efficacy, access to resources, and perceived ability to successfully exit "the life" [4, 26, 27]. Survivors may also experience limited social support and insufficient financial resources to aid them in building a life outside of CSE [26, 27]. These and other barriers clearly point to a need for services that address the individual, social, and environmental barriers that make it difficult for youth survivors to leave and stay out of CSE.

Unfortunately, exiting services and the United States' overall response to CSEY have been poor. In the state of Texas, for example, some state-level legislation fails to focus on affirmative defense provisions, and instead criminalizes persons who were forced, coerced, or threatened to commit crimes while being trafficked [28]. In Las Vegas, Nevada, children as young as 11 years old have been arrested for prostitution, despite being too young to legally consent to having sex [29]. From a human rights perspective, this approach is disempowering and retraumatizing [30].

Practitioners and advocates need evidence-informed approaches that simultaneously a) keep victims safe from harm, b) meet their complex and diverse psychosocial, physical, and spiritual needs in the short-term, and c) build on their internal and external strengths, such that CSEY survivors can grow more resilient to revictimization and re-trafficking over time. Yet, longitudinal research in this area is limited, particularly regarding social work case management services for CSEY survivors [31]. In addition to the challenges researching this population writ large [14, 15], there is ambiguity operationalizing outcomes like "exit" when revictimization is so common [26, 27]. Much like leaving an abusive relationship, it takes many exiting attempts before a survivor can leave CSE completely [26, 27].

Despite these challenges, ecological systems theory [1] and the IEP model [2–4] are promising frameworks practitioners can use to guide social work case management services and conceptualize an "exit" from CSEY. Ecological systems theory demonstrates how the environment influences individual behavior and suggests that interventions ought to be delivered at multiple levels across the social ecology (i.e., micro-, mezzo-, macro-systems). Social work case management services often address how an individual interacts with his or her environment and the systems they occupy [32, 33]. For instance, services might include efforts to improve interpersonal relationships and social support, referrals to community resources, school-based interventions, or advocating for policy change. For CSEY, social work case management may focus on short- and long-term outcomes such as strengthening survivors' bonds with their loved ones, family members, linking them to community institutions, and supporting individuals as they encounter challenges reintegrating to a life without CSE.

The IEP model provides additional specificity on cognitive changes underlying CSE that can inform case management intervention and evaluation. The IEP model [2–4] is an adaptation of a well-known health behavior prediction model, the Integrative Model of Behavioral Prediction (IMBP) [34]. IMBP says the performance of a behavior (i.e., leaving sex trafficking) requires (a) skills to perform the behavior, (b) a lack of environmental constraints prohibiting the behavior, and (c) the intention to perform the behavior [34, 35]. Intention is a key determinant for predicting behavior, and consists of one's attitudes, norms, and efficacy beliefs about the behavior. IMBP states that, if a person has favorable attitudes towards the behavior, experiences social pressure to perform the behavior, and feels efficacious in performing the behavior, they intend to perform the behavior, and will perform the behavior if they have the skills and lack environmental constraints in doing so. The IEP model identifies conditions specific to CSE that may predict an exit, as depicted in Fig 1 below. While the IEP model was originally developed for adult women engaged in street-based prostitution, it has been adapted and tested with commercially sexually exploited youth [36], and can be used with CSEY survivors.

The IEP model finds that intentions to exit include (a) one's attitudes regarding perceived risks and benefits of CSE (risk-recognition and glamorization), (b) confidence in one's ability to resist difficult situations, and (c) resilient beliefs about one's self-worth. It also accounts for (d) diminished agency to exit from barriers like pimps, drug addiction, and more. A benefit of using the IEP model is that it operationalizes cognitive changes or outcomes in the short-term that are a proxy for eventual exit, which researchers can use to evaluate service efficacy. The IEP model has been characterized as an assessment to measure readiness to exit.

For the purposes of this study, we specifically drew social support and case management variables from ecological systems theory, and coping self-efficacy and behavioral change variables from the IEP model. We use these variables, in conjunction with demographic,

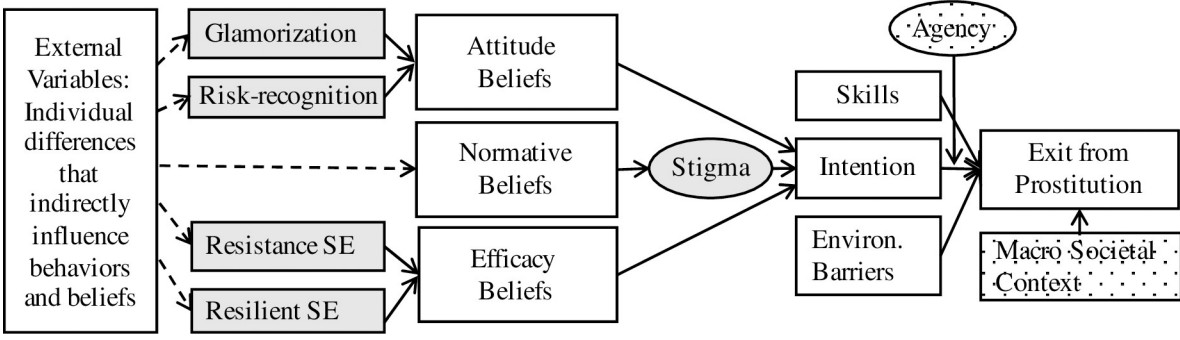

**Fig 1. Intentions to Exit Prostitution (IEP) model, reprinted [4].**

victimization, and service variables, to investigate whether and how social work case management services are associated with positive changes in these outcome variables. The specific research questions (RQs) addressed in this study are:

1. Does social work case management increase CSEY survivors' perceived social support and coping self-efficacy? and

2. Does increased social support and coping self-efficacy predict improvements in CSEY survivors' readiness to change behaviors that may lead to re-exploitation?

This study addresses a sizable gap in the literature on the efficacy of social work case management services on exiting and CSEY survivors' resilience to revictimization and re-trafficking post-exit.

## Materials and methods

This study was designed according to the protocol published by Twis [37]. Although the final protocol described in this manuscript varied slightly from the published protocol (i.e., survey data collected at two points rather than three, timing of study, and nonparametric tests added to the statistical analysis section of the protocol), the study site, design, instruments, and analysis were otherwise carried out as planned. Here we describe the materials and methods for this study; for more detail, please see the published protocol [37].

### Study site and sample

This study used secondary data drawn from a sample of CSEY survivors receiving social work case management services from one agency located in North Texas. Each year, the agency serves hundreds of vulnerable youth. To receive services, youth must be confirmed CSEY survivors (i.e. law enforcement or another official attests to their sex trafficking) or at high-risk for CSEY/trafficking victimization as determined by a validated risk assessment, the Commercial Sexual Exploitation–Identification Tool (CSE-IT) [38]. Social work case management activities include 24/7 crisis response services, client advocacy, goal planning, referral services, and mentorship. Additionally, the agency helps coordinate services with multi-disciplinary team (MDT) partners like juvenile justice, child advocacy centers, CPS, law enforcement, and other community partners. Social work case managers advocate for survivors' needs, ensure they do not fall through service gaps, and emergently respond to community, law enforcement, or child welfare reports of CSEY throughout the region.

Participants were considered eligible for study inclusion if they were actively seeking case management services during the study time period (i.e., January through October 2021). Currently, the agency serves cisgender girls, although they plan to expand services to boys and gender minorities. A total of 95 individuals who received the agency's social work case management services were included in the current study. See Table 1 for an overview of sample demographics.

### Study design

This study was designed as a one-group, quasi-experimental double pre/posttest design wherein CSEY survivors were surveyed by their social work case managers at two timepoints: zero months after study commencement, and again at six months after study commencement. Data collection began in January 2021 and concluded in October 2021; this 10-month study period allowed additional clients to go through intake and data collection following study commencement in January 2021. Importantly, some of the CSEY survivors had been receiving

**Table 1. CSEY survivor data.** Demographics, Victimization History, and Case Management Services.

| Demographic Variable | n (%) or Mean (SD) |
|---|---|
| Race | |
| African American / Black | 41 (43.6) |
| Latinx | 27 (28.7) |
| Multi-racial | 13 (13.8) |
| White | 10 (10.7) |
| Other | 3 (3.3) |
| Age (as of October 2021) | 16.48 (1.93) |
| Sexual Orientation, n = 92 | |
| Straight | 63 (68.5) |
| Lesbian, Gay or Bisexual | 12 (13.1) |
| Unknown | 17 (18.5) |
| County of Residence | |
| County 1 | 66 (70.2) |
| County 2 | 15 (16.0) |
| County 3 | 7 (7.4) |
| County 4 | 5 (5.3) |
| County 5 | 1 (1.1) |
| Trafficking Classification | |
| Confirmed Trafficking | 58 (61.7) |
| High-Risk | 36 (38.3) |
| Total number of secondary victimizations–Mean (SD) | 0.87 (1.32) |
| % Child sexual abuse | 37 (38.9) |
| % Domestic or family violence | 19 (20.2) |
| % Physical abuse or neglect | 16 (16.8) |
| % Bullying | 13 (13.7) |
| % Teen dating violence | 12 (12.6) |
| % Robbery, stalking, or harassment | 7 (7.4) |
| % Non-custodial kidnapping | 4 (4.2) |
| % Child pornography | 3 (3.2) |
| % Mass violence or genocide | 1 (1.1) |
| % Labor trafficking | 1 (1.1) |
| % Other violence victimization (i.e., hit and run, financial, etc.) | 2 (2.1) |
| Total number of special classifications–Mean (SD) | 0.27 (0.54) |
| % Special classification 1—Experiencing homelessness | 9 (9.5) |
| % Special classification 2—Limited English proficiency | 6 (6.3) |
| % Special classification 3—Physical or cognitive disability | 6 (6.3) |
| % Special classification 4—Immigrant or refugee | 4 (4.2) |
| Number of months since initial referral (as of October 2021) | 20.33 (13.13) |
| Number of months between referral and baseline measure date, n = 92 | 13.25 (12.90) |
| Number of services delivered per CSEY survivor (n = 95) | 92.9 (64.4) |
| Number of goals set per CSEY survivor (n = 95) | 3.46 (2.86) |
| % CSEY survivors with goal plan post-intake (n = 95) | 94 (98.9) |
| % CSEY survivors completing two-thirds of their goal plans (n = 82) | 59 (72.0) |

services prior to January 2021, so the two timepoints should not be understood as baseline and six months, but instead as two discrete points in time between which they received additional case management services. The study was considered non-human subjects research by Texas

Christian University's Institutional Review Board (IRB), and was thus exempt from IRB oversight.

## Study instruments

All study instruments were available in both English and Spanish.

## Demographics

Social work case managers collected standard demographic information from survivors, including age, race/ethnicity, county of residence, gender, sexual orientation, and survivors' secondary victimization history as part of their intake process. Social work case managers also documented special classifications, including homelessness, limited English proficiency, disability, and immigrant or refugee status.

## Goal plan

Social work case managers regularly met with CSEY survivors to develop and update individual development plans, which included the survivors' personalized goals. Personalized goals included safety and security goals, physiological goals, and/or other personal goals like educational, vocational, or relational goals. Goal plans were updated monthly, at which point goals were marked as "met" if the goal was achieved, "discontinued" if the goal was no longer part of their plan, or "in progress" if the survivor was still working to achieve the goal. In our analyses, we used variables related to number of goals set, number of goals met, and percentage of goals met.

## Case notes/service count

Social work case managers also tracked survivors' interactions with program staff and the services they received on a weekly basis. Services included case manager check-ins, referrals to outside services, transportation, safety planning, and more. These data were used in the analysis by developing a service count total–for both the number of services received at first assessments and second assessments–to include within our analyses. We also used case management data to calculate the average length of time that survivors had received services between their initial referrals and when initial measures were taken; this data was translated into a variable titled "Months of Service Time", which was calculated by subtracting the date of intake from the date of initial assessments to obtain a month value for the length of time CSEY survivors had received services prior to their first assessments.

## Perceived social support

CSEY survivors' perceived social support was measured with the Multidimensional Scale of Perceived Social Support (MSPSS) [39]. The MSPSS is a brief research tool designed to measure perceptions of support from three sources: family, friends, and significant others. Social work case managers completed the MSPSS with CSEY survivors at zero months following study commencement and again at six months following study commencement. Scale response options were available on a 7-point Likert scale from 1 = "very strongly disagree" to 7 = "very strongly agree". The items were averaged to obtain a total for each subscale and an overall score. Scores can range from 12 to 84; there is not a cutoff value for this scale.

## Coping self-efficacy

CSEY survivors' self-efficacy was measured with the Coping Self-Efficacy Scale (Cop-SE), a 26-item assessment [40] of confidence coping with changeable and unchangeable life stressors. Respondents can indicate their confidence performing each behavior on an 11-point scale (0 = "cannot do at all" to 10 = "certain can do"). Scores can range from 0 to 260; sum scores are tabulated by adding respondents' values for each question. There is not a cutoff value for this scale.

## Readiness to change behaviors

CSEY survivors' readiness to change risky behaviors that can lead to re-exploitation or re-trafficking was measured with a modified version of the University of Rhode Island Change Assessment (URICA) [41], which was a proxy for survivors' readiness to exit. The 12-item URICA scale was modified by replacing "your problem" with one high-risk behavior commonly associated with re-trafficking (i.e., trading or selling sex, running away from home, sexually engaging with strangers online). For example, one question from the scale was "I've been thinking I might want to quit connecting online with strangers", to which respondents could indicate they "strongly disagree" = 1, "somewhat disagree" = 2, are "undecided" = 3, "somewhat agree" = 4, or "strongly agree" = 5. This Likert scoring on a 5-point scale is uniform for all 12 prompts on the scale. For the modified URICA, scores can range from 12 to 60; there is not a cutoff score for this scale. We followed URICA scoring conventions to get four possible readiness categories from the calculated score: pre-contemplation, contemplation, preparation/action, or maintenance.

## Data management and analysis

All data was collected by agency staff as part of their routine service provision (i.e., the research team did not administer any instruments). Demographics were collected within 1–3 days of enrolling the program, case notes were collected weekly, and goal plans updated monthly. The MSPSS, Cop-SE, and URICA were collected at zero months following study commencement and again at six months following study commencement. Staff entered all data into the agency's online data management program or a Qualtrics survey. De-identified data were transferred to the research team via Excel and were uploaded into SPSS version 28 for secondary analysis.

The data were first examined using descriptive statistics (i.e., mean, standard deviation, medians, counts, percentage, etc.) and chi-square analyses to check for normality, assess assumptions, and check associations between variables. Missing data were handled with maximum likelihood estimation or listwise deletion when there was more than 80% missing.

To answer RQ1 on whether social work case management services increase perceived social support among CSEY survivors, we used Wilcoxon signed rank tests (due to sample size) and two separate linear regression models predicting social support and coping on the basis of social work case management services at zero-months. Using zero-month scores allowed for a large sample size with enough power to detect statistically significant variations in outcome scores because the samples for zero-month survey data were much larger than the sample sizes for six-month survey data.

To answer RQ2 on whether social support and coping self-efficacy at one time point predicted readiness to change behaviors that may lead to re-exploitation, we used multinomial logistic regressions. For these tests, we used zero-month URICA data (as a categorical variable) for the outcome variable, in which the possible outcomes were "pre-contemplation", "contemplation", "preparation/action", and "maintenance".

# Results

## Sample characteristics

Table 1 presents the demographic characteristics of the sample. Although the total sample size for this study was 95, only 94 CSEY survivors had complete demographic and service data stored in the agency's online database, and thus the sample size for most of these analyses is 94 unless stated otherwise.

## Demographics

The majority of the sample ($n = 81$, 86.2%) were racial and/or ethnic minorities, including those who identified as African American or Black ($n = 41$, 43.6%), followed closely by Latinx ($n = 27$, 28.7%), and multiracial survivors ($n = 13$, 13.8%). Their average age as of October 2021 was 16.48 years ($SD = 1.93$). The minimum age was 12 and maximum age was 21. Although most of the CSEY survivors included in the sample identified as straight ($n = 63$, 68.5%), 12 clients identified as a sexual minority (13.1%). Finally, most ($n = 81$, 86.2%) of the CSEY survivors in this sample lived in two neighboring counties in the agency's service area.

## Victimization history

Most of the sample were confirmed CSEY survivors ($n = 58$, 61.7%) and experienced more than one type of violence beyond sex trafficking. Child sexual abuse was the most commonly identified secondary victimization ($n = 37$, 38.9%). Child sexual abuse was followed closely by domestic or family violence ($n = 19$, 20.2%), physical abuse or neglect ($n = 16$, 16.8%), bullying ($n = 13$, 13.7%), and teen dating violence ($n = 12$, 12.6%) as the most common types of secondary victimizations within the sample. Most of the survivors in the sample did not have a special classification noted, and only nine CSEY survivors (9.5%) reported experiencing homelessness.

## Social work case management services

The average length of time study participants received services was nearly two years prior to study conclusion ($M = 20.33$ months, $SD = 13.33$ months). The average length of time CSEY survivors had spent receiving services between their intake and their initial survey assessments was 13.25 months ($SD = 12.90$ months). The average number of services delivered to all clients in the sample was 92.9 social work case management services per client ($SD = 64.4$). Descriptive analyses demonstrated that almost all participants developed a goal plan after their intake ($n = 94$, 98.9%), with an average of 3.46 goals set per survivor ($SD = 2.86$). A majority of survivors ($n = 59$, 72.0%) completed at least two-thirds of their goal plans. The average number of goals completed per survivor was 1.94 goals ($SD = 2.34$).

### RQ 1: Impact on CSEY survivors' social support and self-efficacy

## Repeated measures results

Although most of the scale score averages did increase between zero and six months (see Table 2 for matched samples data), none of these changes were statistically significant (see Table 3).

## Linear regression results

The MSPSS Composite linear regression model using zero-month scores was statistically significant ($F_{(5,84)} = 2.86$, $p = 0.02$) and accounted for 15.0% of the total variation in the outcome. See Table 4 for a summary of this model.

**Table 2. MSPSS and Cop-SE average scores for CSEY survivors.** Zero- and six- months following study commencement, for unmatched samples.

| Scale Name and Sample Size | Median *(IQR)* |
|---|---|
| 0-month MSPSS Family Subscale, *n* = 90 | 5.5 *(2.75)* |
| 6-month MSPSS Family Subscale, *n* = 16 | 5.875 *(1.25)* |
| 0-month MSPSS Friends Subscale, *n* = 90 | 5.5 *(2.5)* |
| 6-month MSPSS Friends Subscale, *n* = 15 | 5.0 *(2.0)* |
| 0-month MSPSS Significant Other Subscale, *n* = 90 | 6.5 *(1.25)* |
| 6-month MSPSS Significant Other Subscale, *n* = 16 | 6.25 *(1.32)* |
| 0-month MSPSS Composite Score, *n* = 90 | 16.63 *(4.31)* |
| 6-month MSPSS Composite Score, *n* = 15 | 17.13 *(3.19)* |
| 0-month Cop-SE Scale Score, *n* = 90 | 175 *(63.5)* |
| 6-month Cop-SE Scale Score, *n* = 18 | 194 *(74)* |

The only significant predictor in this model was survivor age, although county of residence and total count of CSEY survivor goals approached significance within the model. Results showed that for every one-year increase in a CSEY survivor's age, the average MSPSS Composite score decreased by 0.41 points. Younger CSEY survivors appear to have better perceived social support than older CSEY survivors. In interpreting the independent variables that approached significance, CSEY survivors living in outlying counties tended to have higher levels of perceived social support than clients living in the two counties in which most survivors lived. Additionally, CSEY survivors with a higher number of set goals tended to have higher levels of perceived social support: For every goal set, MSPSS Composite scores increased by 0.15 points. The number of social work case management services received by survivors was not a significant predictor in any model.

The Cop-SE scale linear regression model using zero-month scores was also statistically significant ($F_{(3,86)}$ = 4.08, $p$ = 0.009), and accounted for 13% of the variation in the outcome. See Table 5 for a summary of this model.

The only significant predictor in this model was CSEY survivor trafficking classification. When survivors were considered "high risk" rather than "confirmed trafficking" (i.e. their trafficking was not confirmed by an external service provider), their Cop-SE scale scores increased by an average of 0.21 points. When interpreting the independent variables that approached significance, for every additional month of service time, CSEY survivors' scores on the Cop-SE scale increased by 0.17 points. CSEY survivors from minority racial and ethnic backgrounds

**Table 3. Wilcoxon signed rank test results.** MSPSS and Cop-SE Repeated Measures.

| Repeated measures | Scale scores and test statistics | | | | |
|---|---|---|---|---|---|
| | *n* | *Test statistic* | *SE* | *Std. test statistic* | *p* |
| 0-month MSPSS–Family Subscale | 16 | 67.50 | 15.89 | 0.94 | 0.35 |
| 6-month MPSSS–Family Subscale | | | | | |
| 0-month MSPSS–Friends Subscale | 15 | 51.50 | 15.90 | -0.06 | 0.95 |
| 6-month MSPSS–Friends Subscale | | | | | |
| 0-month MSPSS–Sig. Other Subscale | 16 | 37.00 | 12.70 | -0.16 | 0.88 |
| 6-month MSPSS–Sig. Other Subscale | | | | | |
| 0-month MSPSS–Composite Score | 15 | 68.50 | 17.58 | 0.48 | 0.63 |
| 6-month MSPSS–Composite Score | | | - | - | |
| 0-month Coping Self-Efficacy | 18 | 109.00 | 21.12 | 1.54 | 0.12 |
| 6-month Coping Self-Efficacy | | | | | |

**Table 4. Linear regression model.** MSPSS (0-month Composite) as Outcome Variable and Demographics and Services as Predictors, n = 90.

|  | B | SE | Beta | t | Sig. |
|---|---|---|---|---|---|
| Constant | 26.72 | 4.06 |  | 6.57 | 0.00 |
| Months of Service Time | 0.04 | 0.03 | 0.15 | 1.23 | 0.22 |
| Age | -0.75 | 0.23 | -0.41 | -3.22 | 0.002* |
| County | 0.74 | 0.41 | 0.19 | 1.79 | 0.08 |
| Trafficking Classification | -0.82 | 0.78 | -0.11 | -1.05 | 0.30 |
| Total Count of Goals | 0.19 | 0.13 | 0.15 | 1.48 | 0.14 |

** $p < .05$.

tended to score lower on the Cop-SE scale, as well. Again, the number of social work case management services received by survivors was not a significant predictor in any model.

## RQ2: Impact of CSEY survivors' social support and self-efficacy on readiness to change

The total number of survivors who completed the URICA scale was 51. Over half of the sample (n = 29, 56.9%) were in the contemplation stage, followed by about one-third who were in pre-contemplation. Only 7.8% (n = 4) of the sample were in the preparation/action stage and no one was in the maintenance stage. Table 6 below provides a summary of CSEY survivors' URICA results at the zero-month measure.

After experimenting with a variety of multinomial logistic regression models, we discovered a model that was significant ($x^2 = 13.70$, $p = 0.03$) and accounted for a large amount of the variability in URICA scale scores (29.0%). Model results are available in Table 7 and likelihood ratio tests in Table 8.

This multinomial logistic regression model demonstrates that for every one-point increase in CSEY survivors' MSPSS Composite scale scores, they are 43% more likely to be in the "Contemplation" stage of change rather than the "Pre-Contemplation" stage of change. Additionally, for every one-point increase in their Coping Self-Efficacy scale scores, survivors are 2% less likely to be in the "Contemplation" stage of change rather than the "Pre-Contemplation" stage of change.

## Discussion

This study investigated the effect of social work case management services on self-efficacy, coping skills, and readiness to exit among youth survivors of CSE. Overall, results of this study suggest that social work case management *may* affect CSEY survivors' perceived social support and coping self-efficacy; in turn, increased social support and coping self-efficacy *does* predict

**Table 5. Linear regression model.** Cop-SE (0-month Composite) as Outcome Variable and Demographics and Services as Predictors, n = 88.

|  | B | SE | Beta | t | Sig. |
|---|---|---|---|---|---|
| Constant | 211.18 | 18.30 |  | 11.54 | 0.00 |
| Months of Service Time | 0.59 | 0.37 | 0.17 | 1.60 | 0.11 |
| Confirmed Trafficked | -19.66 | 9.90 | -0.21 | -1.99 | 0.05* |
| Race | -5.00 | 3.07 | -0.17 | -1.63 | 0.11 |

** $p < .05$.

**Table 6. URICA measurements at zero months.** Average Scale Scores and Counts for URICA as Categorical Variable, n = 51.

| Continuous or Categorical Variable | N (%) |
|---|---|
| Categorical Variables: URICA Readiness Score | |
| Pre-contemplation | 18 (35.3) |
| Contemplation | 29 (56.9) |
| Preparation/Action | 4 (7.8) |
| Maintenance | 0 (0.0) |

improvements in CSEY survivors' intentions to change behaviors that can lead to re-trafficking. Findings from the multinomial logistic regression analyses demonstrate that CSEY survivors who score higher on the social support scale are more likely to contemplate behavior change than those who score lower on this scale. And though the repeated measures tests did not demonstrate that social work case management leads to statistically significantly increased outcome scores, CSEY survivors' scores on coping self-efficacy and perceived social support did trend upwards the more time they had received services. Months of service time also approached statistical significance on the linear regression model for which coping self-efficacy was the outcome variable. These findings show that while social work case management services may not always lead *directly* and *statistically significantly* to improved client outcomes (at least in the format assessed in this study), improved client outcomes lead *directly* and *statistically significantly* to survivors intending to change the behaviors that may lead to sex trafficking re-victimization.

For this reason, the recommendations that follow are heavily focused on how anti-trafficking programs can tweak their existing social work case management services to intentionally target improvements to coping self-efficacy and perceived social support by adhering more closely to the IEP model. By intentionally targeting services, survivors' positive outcomes are likely to improve, these improvements are likely to lead to intentions to change behavior, and intentions to change behavior are likely to lead to long-term behavioral changes like sex trafficking exit.

## Social work case management services to enhance coping self-efficacy

Targeted social work case management programming to help CSEY survivors build coping self-efficacy could take the form of a) skills classes and psychoeducation, b) structured goal-setting check-ins, and c) solution-focused individual or group counseling.

**Table 7. Multinomial logistic regression model.** Predictors of Categorical URICA Readiness Scale Score with Pre-Contemplation as Reference Category, n = 49.

| | *B* | *SE* | *Wald* | *df* | *p* | *Exp(B)* |
|---|---|---|---|---|---|---|
| *Contemplation* | | | | | | |
| Intercept | -1.88 | 1.69 | 1.24 | 1 | 0.27 | |
| Count–Service Contacts | 0.01 | 0.01 | 1.66 | 1 | 0.20 | 1.008 |
| MSPSS Composite Score | 0.36 | 0.15 | 5.28 | 1 | 0.02* | 1.43 |
| Coping Self Efficacy Score | -0.02 | 0.01 | 3.83 | 1 | 0.05* | 0.98 |
| *Preparation/Action* | | | | | | |
| Intercept | -2.98 | 2.66 | 1.26 | 1 | 0.26 | |
| Count–Service Contacts | -0.02 | 0.01 | 1.88 | 1 | 0.17 | 0.98 |
| MSPSS Composite Score | -0.02 | 0.18 | 0.01 | 1 | 0.92 | 0.98 |
| Coping Self Efficacy Score | 0.02 | 0.02 | 0.97 | 1 | 0.32 | 1.02 |

* $p < .05$

**Table 8. Multinomial logistic regression likelihood ratio tests.** Outcome Variable as Categorical URICA Readiness Score, n = 49.

| | -2 Log Likelihood | $x^2$ | df | p |
|---|---|---|---|---|
| Intercept | 75.90 | 2.22 | 2 | 0.329 |
| Count–Service Contacts | 79.67 | 5.99 | 2 | 0.05* |
| MSPSS Composite Score—Baseline | 81.12 | 7.44 | 2 | 0.02* |
| Coping Self-Efficacy Score–Baseline | 81.10 | 7.43 | 2 | 0.02* |

\* p < .05.

## Skills classes and psychoeducation

According to seminal self-efficacy theorist Albert Bandura [42], the best way to create self-efficacy is to provide individuals with opportunities to master tasks and to expose individuals to similar people who also master tasks. These efforts can be further strengthened through social persuasion, i.e., strengthening individuals' beliefs that they can succeed. While it is likely that social persuasion is a component of social work case management by nature of the relationship between social workers and survivors, skills classes and psychoeducation could provide both task mastery insight and peer modeling to survivors. It is important for survivors to build skills, and to believe in their ability to succeed at these skills, because the IEP model states that performing desired behaviors hinges on an individual's skillset to enact the behavior [2–4]. The skills required to exit CSEY in the long-term include educational skills, job acquisition and retention skills, emotional regulation skills, self-care skills, and social support skills. By teaching these skills in classes or psychoeducation, survivors will have an opportunity to learn new skills and interact with other survivors who are also learning and implementing these skills. Combined with goal-setting and solution-focused group counseling, this targeted social work case management activity is likely to promote coping self-efficacy and longer-term behavioral change.

## Structured goal-setting check-ins

The social work case managers in this study consistently performed goal-setting activities with their clients. With increased precision, this case management activity is likely to enhance coping self-efficacy and longer-term behavior change, as well. Indeed, a large body of research suggests that even small efforts towards goal setting and goal adherence can add up to larger behavioral adaptations. For instance, the authors of one meta-analysis explored the results of 141 randomized controlled trials and found that goal setting, in and of itself, is associated with positive behavioral changes [43]. The authors of this study further found that goal setting was particularly effective for behavioral change when the set goals were difficult, public, and shared by other people [43]. Anti-trafficking agencies that offer social work case management services should consider tightening their processes and procedures related to survivor goal setting and follow up. Regular goal-setting conversations with survivors will likely improve actual goal adherence, for both small goals and large goals like sex trafficking exit. Agencies might also consider implementing a rewards system for goal setting and adherence. Doing so, even if goals remain anonymous within the agency to reduce competition, might make goal-setting and adherence more public and communal, thereby enhancing perceived social support as well.

## Solution-focused group counseling

One final recommendation for improving survivors' coping self-efficacy is the implementation of group-based solution focused brief therapy (SFBT). Numerous research studies point to

group SFBT as helpful for building self-efficacy among adolescents [44]. Group SFBT can help adolescents reflect on their lives, observe their peers' change efforts, and plan for the positive outcomes that they articulate for themselves through goal-setting and follow-up with their social work case managers and again in the group setting. Group SFBT also creates "a context for sharing, providing and receiving social support, encouragement and empowerment, which are important elements in the process of self-efficacy strengthening," [44].

### Social work case management services to build social support

Anti-trafficking agencies should also consider adding social support programming to their case management services, especially peer support programming, in order to more clearly build social support in CSEY survivors. This recommended emphasis on peer support is rooted in the finding that survivors' perceived social support from friends actually trends downward over time. This makes logical sense, since youth who leave sex trafficking are likely to lose touch with the friends they had while in "the life". Anti-trafficking agencies should, therefore, offer additional opportunities for clients to build networks and social support with new friends in a way that makes sense for the service delivery context. Social support programming could look like group outings or peer support groups, or it could take the form of a more structured approach to social support and group work. In keeping with previous recommendations to more intentionally target programming, it may be wise for anti-trafficking agencies to adopt manualized interventions that are likely to build social support, and perhaps coping self-efficacy and task mastery, as well. Beyond SFBT, other examples include Interpersonal Therapy (IPT) for Groups, which is an evidence-based intervention that can be used to treat a range of client populations. The approach helps clients work on "interpersonal problems and. . .develop interpersonal skills with other [clients] struggling with similar challenges" [45, 46]. Other group formats could include Dialectical Behavioral Therapy (DBT) skills groups, or structured psychoeducation.

### Limitations

Despite the significant efforts that we made to enhance the credibility of this study, several methodological weaknesses could potentially limit the validity and reliability of the results. The limitations of this study include its small sample size (particularly for repeated measures tests), its geographically restricted sample, and its potential for data collection bias. The small sample size increased the risk of type II statistical errors, meaning that we may have accepted the null hypothesis when it was actually false (i.e., false negative). Additionally, the sample was restricted to CSEY survivors residing in North Texas, so the generalizability of this study's findings cannot be assumed for other geographic regions beyond the agency's service area. Additionally, many CSEY survivors had received services for months before study commencement, which may skew the findings. Finally, it is possible that the study's reliance on social work case managers and program staff for data collection could have biased or skewed the results. When social work case managers collect data from their own clients, scores are at-risk for artificial inflation from both staff and CSEY survivors. Given these limitations, study results should be considered preliminary and warrant further investigation. Replication studies–especially those with larger samples–should be considered for future evaluation efforts.

### Conclusions

This study represented a unique effort to assess social work case management interventions with CSEY survivors. Study results point results point towards the utility of the ecological systems theory and the IEP model as frameworks for both practice and efficacy research with

CSEY survivors. Social work case management with CSEY survivors should center on the IEP model, and encourage client improvements in coping self-efficacy and social support in order to enhance behavioral changes that may lead to long-term sex trafficking exit. Future research should build on study results by replicating data collection and statistical analyses with larger sample sizes, and perhaps additional control or comparison groups. Future research should also focus on collecting primary data from survivors in an attempt to replicate study findings.

## Supporting information

**S1 File. Readiness to change surveys.** URICA measures in English and Spanish.
(DOCX)

**S2 File. Social support surveys.** MSPSS measure in English and Spanish.
(DOCX)

**S3 File. Self-efficacy surveys.** Coping self-efficacy measure in English and Spanish.
(DOCX)

**S4 File. Minimal data set.** De-identified data set containing underlying data to support the study's conclusions.
(SAV)

## Acknowledgments

This study was made possible by logistical support from Traffick911.

## Author Contributions

**Conceptualization:** Mary K. Twis, Andrea Cimino.

**Data curation:** Mary K. Twis, Marilyn Plunk.

**Formal analysis:** Mary K. Twis, Andrea Cimino.

**Funding acquisition:** Mary K. Twis.

**Investigation:** Mary K. Twis.

**Methodology:** Mary K. Twis, Andrea Cimino.

**Project administration:** Mary K. Twis.

**Writing – original draft:** Mary K. Twis.

**Writing – review & editing:** Mary K. Twis, Andrea Cimino, Marilyn Plunk.

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
