## [Decision Letter · Decision Letter 0]

16 Aug 2023

PONE-D-23-02821Coping Self- Efficacy and Social Support as Predictors of Adolescent Sex Trafficking Exit: Results of a Secondary AnalysisPLOS ONE

Dear Dr. Twis,

Thank you for submitting your manuscript to PLOS ONE. After careful consideration, we feel that it has merit but does not fully meet PLOS ONE’s publication criteria as it currently stands. Therefore, we invite you to submit a revised version of the manuscript that addresses the points raised during the review process.

Accepted with minor changes:

Reviewer 01

In introduction section, give some statistics of victims/cases/incidents of the problem (if any) globally as well as locally where the study was conducted in order to clearly understand the severity of the problem.

Reviewer 02

Good effort and you have presented all the data in proper way. It is very informative but if you conduct this study by taking the primary data it would more beneficial. I suggest you can conduct primary study on this topic in future.

We look forward to receiving your revised manuscript.

Kind regards,

Sadia Jabeen, Ph.D.

Academic Editor

PLOS ONE

Journal Requirements:

This study was made possible by the support of the Office of the Texas Governor’s Child Sex Trafficking Team and Traffick911.

However, funding information should not appear in the Acknowledgments section or other areas of your manuscript. We will only publish funding information present in the Funding Statement section of the online submission form. 

This project was supported by a Victims of Crime Act (VOCA) federal pass-through grant awarded to the Office of the Texas Governor (OOG) Child Sex Trafficking Team and Traffick911. The sponsor of this protocol – Traffick911 – has assisted with the study design and data collection plan described in this protocol. Traffick911 administrators have approved of the protocol described herein. The original funders of this project, including VOCA and the OOG, had no role in study design, data collection and analysis, decision to publish, or preparation of the manuscript. The opinions, findings, conclusions, and recommendations expressed in this publication are those of the authors and do not necessarily reflect the views of VOCA, OOG, or Traffick911. (www.traffick911.com).

Additional Editor Comments:

Accepted with minor changes:

Reviewer 01

In introduction section, give some statistics of victims/cases/incidents of the problem (if any) globally as well as locally where the study was conducted in order to clearly understand the severity of the problem.

Reviewer 02

Good effort and you have presented all the data in proper way. It is very informative but if you conduct this study by taking the primary data it would more beneficial. I suggest you can conduct primary study on this topic in future.

Reviewers' comments:

Reviewer's Responses to Questions

**Comments to the Author**

1. Is the manuscript technically sound, and do the data support the conclusions?

Reviewer #1: Yes

Reviewer #2: Yes

2. Has the statistical analysis been performed appropriately and rigorously? 

Reviewer #1: Yes

Reviewer #2: Yes

3. Have the authors made all data underlying the findings in their manuscript fully available?

Reviewer #1: Yes

Reviewer #2: Yes

4. Is the manuscript presented in an intelligible fashion and written in standard English?

Reviewer #1: Yes

Reviewer #2: Yes

5. Review Comments to the Author

Reviewer #1: In introduction section, give some statistics of victims/cases/incidents of the problem (if any) globally as well as locally where the study was conducted in order to clearly understand the severity of the problem.

Reviewer #2: Good effort and you have presented all the data in proper way. It is very informative but if you conduct this study by taking the primary data it would more beneficial. I suggest you can conduct primary study on this topic in future.

6. PLOS authors have the option to publish the peer review history of their article (what does this mean?). If published, this will include your full peer review and any attached files.

Reviewer #1: No

Reviewer #2: No

---

## [Author Response · Author response to Decision Letter 0]

23 Aug 2023

Please see the included document "Response to Reviewers" for a point-by-point response to the reviewers and editors.

---

## [Editor Report · Decision Letter 1]

24 Aug 2023

Coping Self- Efficacy and Social Support as Predictors of Adolescent Sex Trafficking Exit: Results of a Secondary Analysis

PONE-D-23-02821R1

Dear Dr. Twis,

We’re pleased to inform you that your manuscript has been judged scientifically suitable for publication and will be formally accepted for publication once it meets all outstanding technical requirements.

Kind regards,

Sadia Jabeen, Ph.D.

Academic Editor

PLOS ONE
---

## [Editor Report · Acceptance letter]

31 Aug 2023

PONE-D-23-02821R1 

Coping Self- Efficacy and Social Support as Predictors of Adolescent Sex Trafficking Exit: Results of a Secondary Analysis 

Dear Dr. Twis:

I'm pleased to inform you that your manuscript has been deemed suitable for publication in PLOS ONE. Congratulations! Your manuscript is now with our production department. 

Kind regards, 

on behalf of

Dr. Sadia Jabeen 

Academic Editor

PLOS ONE